# GAN Dissection:
# Visualizing and Understanding
# Generative Adversarial Networks

**David Bau[1,2], Jun-Yan Zhu[1], Hendrik Strobelt[2,3], Bolei Zhou[4],**
**Joshua B. Tenenbaum[1], William T. Freeman[1], Antonio Torralba[1,2]**
[1]Massachusetts Institute of Technology, [2]MIT-IBM Watson AI Lab,
[3]IBM Research, [4]The Chinese University of Hong Kong

## Abstract

Generative Adversarial Networks (GANs) have recently achieved impressive results for many real-world applications, and many GAN variants have emerged with improvements in sample quality and training stability. However, they have not been well visualized or understood. How does a GAN represent our visual world internally? What causes the artifacts in GAN results? How do architectural choices affect GAN learning? Answering such questions could enable us to develop new insights and better models.

In this work, we present an analytic framework to visualize and understand GANs at the unit-, object-, and scene-level. We first identify a group of interpretable units that are closely related to object concepts using a segmentation-based network dissection method. Then, we quantify the causal effect of interpretable units by measuring the ability of interventions to control objects in the output. We examine the contextual relationship between these units and their surroundings by inserting the discovered object concepts into new images. We show several practical applications enabled by our framework, from comparing internal representations across different layers, models, and datasets, to improving GANs by locating and removing artifact-causing units, to interactively manipulating objects in a scene. We provide open source interpretation tools to help researchers and practitioners better understand their GAN models.

## 1 Introduction

Generative Adversarial Networks (GANs) (Goodfellow et al., 2014) have been able to produce photorealistic images, often indistinguishable from real images. This remarkable ability has powered many real-world applications ranging from visual recognition (Wang et al., 2017), to image manipulation (Isola et al., 2017; Zhu et al., 2017), to video prediction (Mathieu et al., 2016). Since their invention in 2014, many GAN variants have been proposed (Radford et al., 2016; Zhang et al., 2018a), often producing more realistic and diverse samples with better training stability.

Despite this tremendous success, many questions remain to be answered. For example, to produce a church image (Figure 1a), what knowledge does a GAN need to learn? Alternatively, when a GAN sometimes produces terribly unrealistic images (Figure 1f), what causes the mistakes? Why does one GAN variant work better than another? What fundamental differences are encoded in their weights?

In this work, we study the internal representations of GANs. To a human observer, a well-trained GAN appears to have learned facts about the objects in the image: for example, a door can appear on a building but not on a tree. We wish to understand how a GAN represents such structure. Do the objects emerge as pure pixel patterns without any explicit representation of objects such as doors and trees, or does the GAN contain internal variables that correspond to the objects that humans perceive? If the GAN does contain variables for doors and trees, do those variables cause the generation of those objects, or do they merely correlate? How are relationships between objects represented?

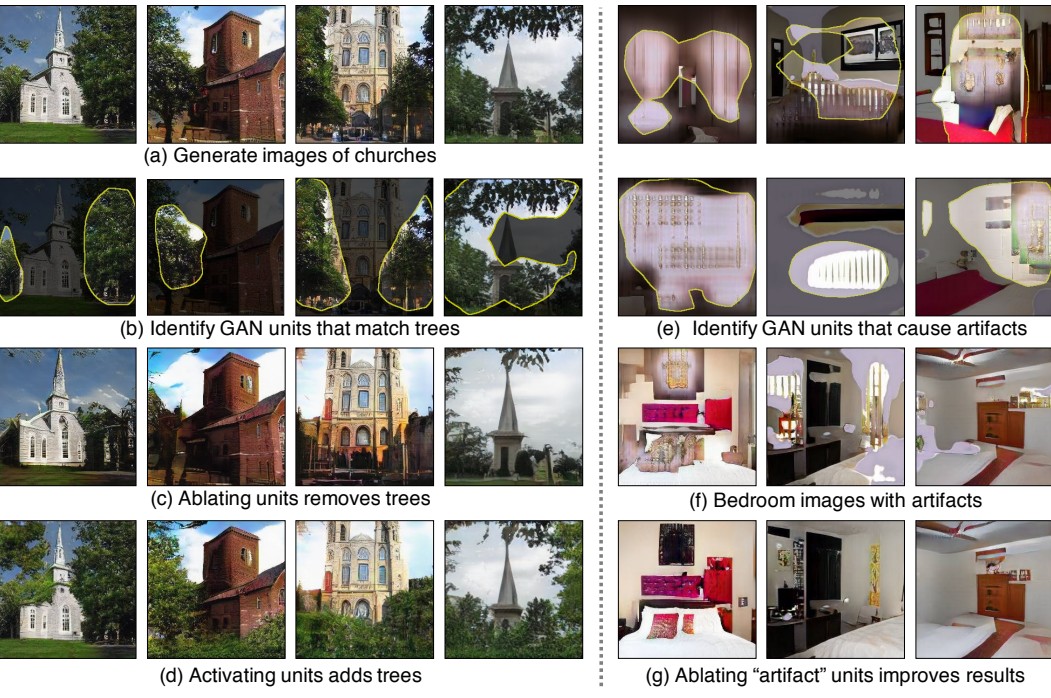

Figure 1: Overview: (a) Realistic outdoor church images generated by Progressive GANs (Karras et al., 2018). (b) Given a pre-trained GAN model, we identify a set of interpretable units whose featuremap is correlated to an object class across different images. For example, one unit in `layer4` localizes tree regions with diverse visual appearance. (c) We force the activation of the units to be zero and quantify the average casual effect of the ablation. Here we successfully remove trees from church images. (d) We activate tree causal units in other locations. These same units synthesize new trees, visually compatible with their surrounding context. In addition, our method can diagnose and improve GANs by identifying artifact-causing units (e). We can remove the artifacts that appear (f) and significantly improve the results by ablating the "artifact" units (g). Please see our demo video.

We present a general method for visualizing and understanding GANs at different levels of abstraction, from each neuron, to each object, to the contextual relationship between different objects. We first identify a group of interpretable units that are related to object concepts (Figure 1b). These units' featuremaps closely match the semantic segmentation of a particular object class (e.g., trees). Second, we directly intervene within the network to identify sets of units that cause a type of objects to disappear (Figure 1c) or appear (Figure 1d). We quantify the causal effect of these units using a standard causality metric. Finally, we examine the contextual relationship between these causal object units and the background. We study where we can insert object concepts in new images and how this intervention interacts with other objects in the image (Figure 1d). To our knowledge, our work provides the first systematic analysis for understanding the internal representations of GANs.

Finally, we show several practical applications enabled by this analytic framework, from comparing internal representations across different layers, GAN variants and datasets; to debugging and improving GANs by locating and ablating "artifact" units (Figure 1e); to understanding contextual relationships between objects in scenes; to manipulating images with interactive object-level control.

## 2 RELATED WORK

**Generative Adversarial Networks.** The quality and diversity of results from GANs (Goodfellow et al., 2014) has continued to improve, from generating simple digits and faces (Goodfellow et al., 2014), to synthesizing natural scene images (Radford et al., 2016; Denton et al., 2015), to generating 1k photorealistic portraits (Karras et al., 2018), to producing one thousand object classes (Miyato et al., 2018; Zhang et al., 2018a). GANs have also enabled applications such as visual recognition (Wang et al., 2017; Hoffman et al., 2018), image manipulation (Isola et al., 2017; Zhu et al., 2017), and video generation (Mathieu et al., 2016; Wang et al., 2018). Despite the successes, little work has been done to visualize what GANs have learned. Prior work (Radford et al., 2016; Zhu et al., 2016; Brock et al., 2017) manipulates latent vectors and observes how the results change accordingly.

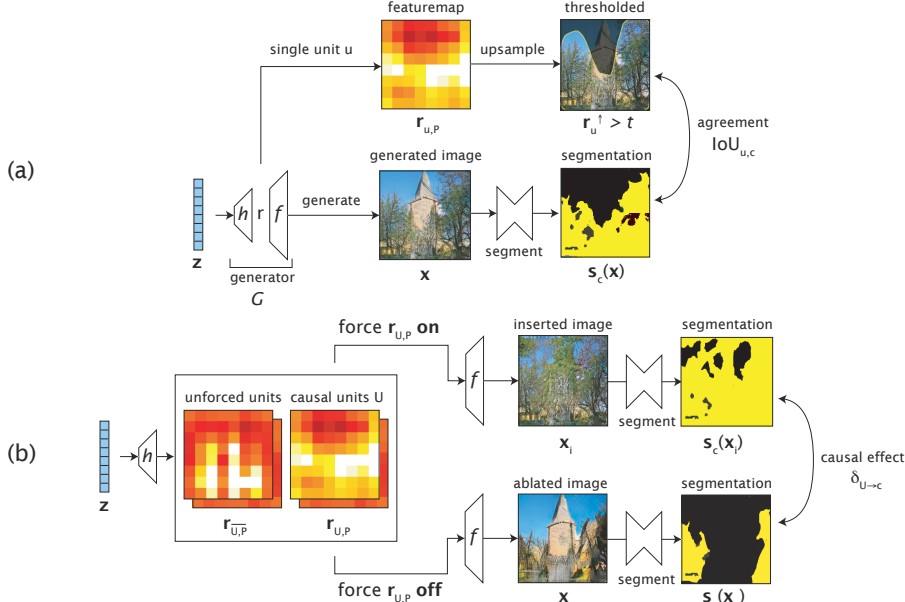

Figure 2: Measuring the relationship between representation units and trees in the output using (a) dissection and (b) intervention. Dissection measures agreement between a unit $u$ and a concept $c$ by comparing its thresholded upsampled heatmap with a semantic segmentation of the generated image $s_c(x)$. Intervention measures the causal effect of a set of units $U$ on a concept $c$ by comparing the effect of forcing these units on (unit insertion) and off (unit ablation). The segmentation $s_c$ reveals that trees increase after insertion and decrease after ablation. The average difference in the tree pixels measures the average causal effect. In this figure, interventions are applied to the entire featuremap $\mathbb{P}$, but insertions and ablations can also apply to any subset of pixels $\mathrm{P} \subset \mathbb{P}$.

**Visualizing deep neural networks.** A CNN can be visualized by reconstructing salient image features (Simonyan et al., 2014; Mahendran & Vedaldi, 2015) or by mining patches that maximize hidden layers' activations (Zeiler & Fergus, 2014); or we can synthesize input images to invert a feature layer (Dosovitskiy & Brox, 2016). Alternately, we can identify the semantics of each unit (Zhou et al., 2015; Bau et al., 2017; Zhou et al., 2018a) by measuring agreement between unit activations and object segmentation masks, or by training a network to increase interpretability of such units (Zhang et al., 2018b). Visualization of an RNN has also revealed interpretable units that track long-range dependencies (Karpathy et al., 2016; Strobelt et al., 2018). Most previous work on network visualization has focused on networks trained for classification; our work explores deep generative models trained for image generation.

**Understanding neural representation in biology.** Studies of biological neural networks find evidence of both local representations in which individual neurons are selective for meaningful concepts (Quiroga, 2012), as well as distributed representations in which individual neurons are essentially meaningless (Yuste, 2015). Computational models of biological learning (Bowers et al., 2016; Dasgupta et al., 2018) find sparse and local representations can aid generalization to novel stimuli.

**Explaining the decisions of deep neural networks.** Individual network decisions can be explained using informative heatmaps (Zhou et al., 2018b; 2016; Selvaraju et al., 2017) or by scoring salience (Simonyan et al., 2014; Bach et al., 2015; Sundararajan et al., 2017; Lundberg & Lee, 2017). Such analyses reveals which inputs contribute most to a categorical prediction by a network. Recent work has also studied the contribution of feature vectors (Kim et al., 2017; Zhou et al., 2018b) or individual channels (Olah et al., 2018) to a final prediction, and Morcos et al. (2018) has examined the effect of individual units by ablating them. Those methods explain discriminative classifiers. Our method aims to explain how an image can be generated by a network, which is much less explored.

## 3 METHOD

Our goal is to analyze how objects such as trees are encoded by the internal representations of a GAN generator $G: \mathbf{z} \to \mathbf{x}$. Here $\mathbf{z} \in \mathbb{R}^{|z|}$ denotes a latent vector sampled from a low-dimensional distribution, and $\mathbf{x} \in \mathbb{R}^{H \times W \times 3}$ denotes an $H \times W$ generated image. We use *representation* to

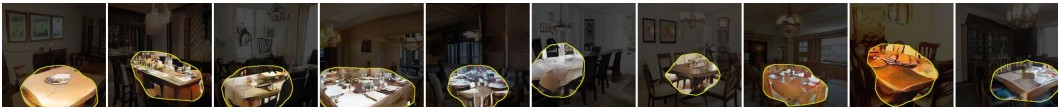

Thresholding unit #65 layer 3 of a dining room generator matches 'table' segmentations with IoU=0.34.

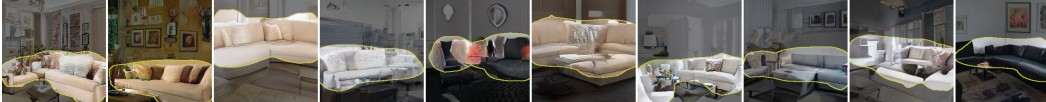

Thresholding unit #37 layer 4 of a living room generator matches 'sofa' segmentations with IoU=0.29.

Figure 3: Visualizing the activations of individual units in two GANs. Top ten activating images are shown, and IoU is measured over a sample of 1000 images. In each image, the unit feature is upsampled and thresholded as described in Eqn. 2.

describe the tensor $\mathbf{r}$ output from a particular layer of the generator $G$, where the generator creates an image $\mathbf{x}$ from random $\mathbf{z}$ through a composition of layers: $\mathbf{r} = h(\mathbf{z})$ and $\mathbf{x} = f(\mathbf{r}) = f(h(\mathbf{z})) = G(\mathbf{z})$.

Since $\mathbf{r}$ has all the data necessary to produce the image $\mathbf{x} = f(\mathbf{r})$, $\mathbf{r}$ certainly contains the information to deduce the presence of any visible class $c$ in the image. Therefore the question we ask is not whether information about $c$ is present in $\mathbf{r}$ — it is — but *how* such information is encoded in $\mathbf{r}$. In particular, for any class from a universe of concepts $c \in \mathbb{C}$, we seek to understand whether $\mathbf{r}$ explicitly represents $c$ in some way where it is possible to factor $\mathbf{r}$ at locations P into two components

$$\mathbf{r}_{\mathbb{U},\mathrm{P}} = (\mathbf{r}_{\mathrm{U},\mathrm{P}}, \mathbf{r}_{\overline{\mathrm{U}},\mathrm{P}}), \tag{1}$$

where the generation of the object $c$ at locations P depends mainly on the units $\mathbf{r}_{\mathrm{U},\mathrm{P}}$, and is insensitive to the other units $\mathbf{r}_{\overline{\mathrm{U}},\mathrm{P}}$. Here we refer to each channel of the featuremap as a unit: U denotes the set of unit indices of interest and $\overline{\mathrm{U}}$ is its complement; we will write $\mathbb{U}$ and $\mathbb{P}$ to refer to the entire set of units and featuremap pixels in $r$. We study the structure of $\mathbf{r}$ in two phases:

- Dissection: starting with a large dictionary of object classes, we identify the classes that have an explicit representation in $\mathbf{r}$ by measuring the agreement between individual units of $\mathbf{r}$ and every class $c$ (Figure 1b).
- Intervention: for the represented classes identified through dissection, we identify causal sets of units and measure causal effects between units and object classes by forcing sets of units on and off (Figure 1c,d).

## 3.1 Characterizing units by dissection

We first focus on individual units of the representation. Recall that $\mathbf{r}_{u,\mathbb{P}}$ is the one-channel $h \times w$ featuremap of unit $u$ in a convolutional generator, where $h \times w$ is typically smaller than the image size. We want to know if a specific unit $\mathbf{r}_{u,\mathbb{P}}$ encodes a semantic class such as a "tree". For image classification networks, Bau et al. (2017) has observed that many units can approximately locate emergent object classes when the units are upsampled and thresholded. In that spirit, we select a universe of concepts $c \in \mathbb{C}$ for which we have a semantic segmentation $\mathbf{s}_c(\mathbf{x})$ for each class. Then we quantify the spatial agreement between the unit $u$'s thresholded featuremap and a concept $c$'s segmentation with the following intersection-over-union (IoU) measure:

$$\mathrm{IoU}_{u,c} \equiv \frac{\mathbb{E}_{\mathbf{z}} \left| (\mathbf{r}^{\uparrow}_{u,\mathbb{P}} > t_{u,c}) \wedge \mathbf{s}_c(\mathbf{x}) \right|}{\mathbb{E}_{\mathbf{z}} \left| (\mathbf{r}^{\uparrow}_{u,\mathbb{P}} > t_{u,c}) \vee \mathbf{s}_c(\mathbf{x}) \right|}, \text{ where } t_{u,c} = \arg\max_{t} \frac{\mathrm{I}(\mathbf{r}^{\uparrow}_{u,\mathbb{P}} > t; \mathbf{s}_c(\mathbf{x}))}{\mathrm{H}(\mathbf{r}^{\uparrow}_{u,\mathbb{P}} > t, \mathbf{s}_c(\mathbf{x}))}, \tag{2}$$

where $\wedge$ and $\vee$ denote intersection and union operations, and $\mathbf{x} = G(\mathbf{z})$ denotes the image generated from $\mathbf{z}$. The one-channel feature map $\mathbf{r}_{u,\mathbb{P}}$ slices the entire featuremap $\mathbf{r} = h(\mathbf{z})$ at unit $u$. As shown in Figure 2a, we upsample $\mathbf{r}_{u,\mathbb{P}}$ to the output image resolution as $\mathbf{r}^{\uparrow}_{u,\mathbb{P}}$. $(\mathbf{r}^{\uparrow}_{u,\mathbb{P}} > t_{u,c})$ produces a binary mask by thresholding the $\mathbf{r}^{\uparrow}_{u,\mathbb{P}}$ at a fixed level $t_{u,c}$. $\mathbf{s}_c(\mathbf{x})$ is a binary mask where each pixel indicates the presence of class $c$ in the generated image $\mathbf{x}$. The threshold $t_{u,c}$ is chosen to be informative as possible by maximizing the information quality ratio I/H (using a separate validation set), that is, it maximizes the portion of the joint entropy H which is mutual information I (Wijaya et al., 2017).

We can use $\mathrm{IoU}_{u,c}$ to rank the concepts related to each unit and label each unit with the concept that matches it best. Figure 3 shows examples of interpretable units with high $\mathrm{IoU}_{u,c}$. They are not the

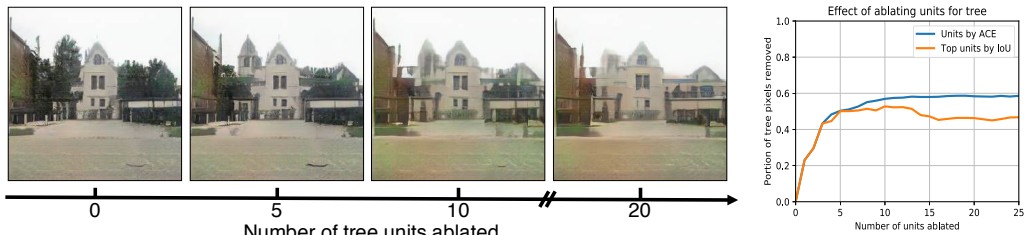

Figure 4: Ablating successively larger sets of tree-causal units from a GAN trained on LSUN outdoor church images, showing that the more units are removed, the more trees are reduced, while buildings remain. The choice of units to ablate is specific to the tree class and does not depend on the image. At right, the causal effect of removing successively more tree units is plotted, comparing units chosen to optimize the average causal effect (ACE) and units chosen with the highest IoU for trees.

only units to match tables and sofas: `layer3` of the dining room generator has 31 units (of 512) that match tables and table parts, and `layer4` of the living room generator has 65 (of 512) sofa units.

Once we have identified an object class that a set of units match closely, we next ask: which units are responsible for triggering the rendering of that object? A unit that correlates highly with an output object might not actually cause that output. Furthermore, any output will jointly depend on several parts of the representation. We need a way to identify combinations of units that cause an object.

## 3.2 MEASURING CAUSAL RELATIONSHIPS USING INTERVENTION

To answer the above question about causality, we probe the network using interventions: we test whether a set of units U in $\mathbf{r}$ cause the generation of $c$ by forcing the units of U on and off.

Recall that $\mathbf{r}_{\mathrm{U,P}}$ denotes the featuremap $\mathbf{r}$ at units U and locations P. We *ablate* those units by forcing $\mathbf{r}_{\mathrm{U,P}} = \mathbf{0}$. Similarly, we *insert* those units by forcing $\mathbf{r}_{\mathrm{U,P}} = \mathbf{k}$, where $\mathbf{k}$ is a per-class constant, as described in Section S-6.4. We decompose the featuremap $\mathbf{r}$ into two parts $(\mathbf{r}_{\mathrm{U,P}}, \mathbf{r}_{\overline{\mathrm{U,P}}})$, where $\mathbf{r}_{\overline{\mathrm{U,P}}}$ are unforced components of $\mathbf{r}$:

$$\text{Original image}: \quad \mathbf{x} = G(\mathbf{z}) \equiv f(\mathbf{r}) \equiv f(\mathbf{r}_{\mathrm{U,P}}, \mathbf{r}_{\overline{\mathrm{U,P}}}) \tag{3}$$

$$\text{Image with U ablated at pixels P}: \quad \mathbf{x}_a = f(\mathbf{0}, \mathbf{r}_{\overline{\mathrm{U,P}}})$$

$$\text{Image with U inserted at pixels P}: \quad \mathbf{x}_i = f(\mathbf{k}, \mathbf{r}_{\overline{\mathrm{U,P}}})$$

An object is caused by U if the object appears in $\mathbf{x}_i$ and disappears from $\mathbf{x}_a$. Figure 1c demonstrates the ablation of units that remove trees, and Figure 1d demonstrates insertion of units at specific locations to make trees appear. This causality can be quantified by comparing the presence of trees in $\mathbf{x}_i$ and $\mathbf{x}_a$ and averaging effects over all locations and images. Following prior work (Holland, 1988; Pearl, 2009), we define the average causal effect (ACE) of units U on the generation of on class $c$ as:

$$\delta_{\mathrm{U} \to c} \equiv \mathbb{E}_{\mathbf{z},\mathrm{P}}[\mathbf{s}_c(\mathbf{x}_i)] - \mathbb{E}_{\mathbf{z},\mathrm{P}}[\mathbf{s}_c(\mathbf{x}_a)], \tag{4}$$

where $\mathbf{s}_c(\mathbf{x})$ denotes a segmentation indicating the presence of class $c$ in the image $\mathbf{x}$ at P. To permit comparisons of $\delta_{\mathrm{U} \to c}$ between classes $c$ which are rare, we normalize our segmentation $\mathbf{s}_c$ by $\mathbb{E}_{\mathbf{z},\mathrm{P}}[\mathbf{s}_c(x)]$. While these measures can be applied to a single unit, we have found that objects tend to depend on more than one unit. Thus we wish to identify a set of units U that maximize the average causal effect $\delta_{\mathrm{U} \to c}$ for an object class $c$.

**Finding sets of units with high ACE.** Given a representation $\mathbf{r}$ with $d$ units, exhaustively searching for a fixed-size set U with high $\delta_{\mathrm{U} \to c}$ is prohibitive as it has $\binom{d}{|\mathrm{U}|}$ subsets. Instead, we optimize a continuous intervention $\boldsymbol{\alpha} \in [0,1]^d$, where each dimension $\boldsymbol{\alpha}_u$ indicates the degree of intervention for a unit $u$. We maximize the following average causal effect formulation $\delta_{\boldsymbol{\alpha} \to c}$:

$$\text{Image with partial ablation at pixels P}: \quad \mathbf{x}'_a = f((\mathbf{1} - \boldsymbol{\alpha}) \odot \mathbf{r}_{\mathbb{U},\mathrm{P}}, \ \mathbf{r}_{\mathbb{U},\overline{\mathrm{P}}}) \tag{5}$$

$$\text{Image with partial insertion at pixels P}: \quad \mathbf{x}'_i = f(\boldsymbol{\alpha} \odot \mathbf{k} + (\mathbf{1} - \boldsymbol{\alpha}) \odot \mathbf{r}_{\mathbb{U},\mathrm{P}}, \ \mathbf{r}_{\mathbb{U},\overline{\mathrm{P}}})$$

$$\text{Objective}: \quad \delta_{\boldsymbol{\alpha} \to c} = \mathbb{E}_{\mathbf{z},\mathrm{P}}[\mathbf{s}_c(\mathbf{x}'_i)] - \mathbb{E}_{\mathbf{z},\mathrm{P}}[\mathbf{s}_c(\mathbf{x}'_a)],$$

where $\mathbf{r}_{\mathbb{U},\mathrm{P}}$ denotes the all-channel featuremap at locations P, $\mathbf{r}_{\mathbb{U},\overline{\mathrm{P}}}$ denotes the all-channel featuremap at other locations $\overline{\mathrm{P}}$, and $\odot$ applies a per-channel scaling vector $\boldsymbol{\alpha}$ to the featuremap $\mathbf{r}_{\mathbb{U},\mathrm{P}}$. We optimize

$\boldsymbol{\alpha}$ over the following loss with an L2 regularization:

$$\boldsymbol{\alpha}^* = \arg\min_{\boldsymbol{\alpha}}(-\delta_{\boldsymbol{\alpha} \to c} + \lambda||\boldsymbol{\alpha}||_2), \quad (6)$$

where $\lambda$ controls the relative importance of each term. We add the L2 loss as we seek a minimal set of casual units. We optimize using stochastic gradient descent, sampling over both $\mathbf{z}$ and featuremap locations P and clamping the coefficient $\boldsymbol{\alpha}$ within the range $[0, 1]^d$ at each step (d is the total number of units). More details of this optimization are discussed in Section S-6.4. Finally, we can rank units by $\boldsymbol{\alpha}_u^*$ and achieve a stronger causal effect (i.e., removing trees) when ablating successively larger sets of tree-causing units as shown in Figure 4.

## 4 RESULTS

We study three variants of Progressive GANs (Karras et al., 2018) trained on LSUN scene datasets (Yu et al., 2015). To segment the generated images, we use a recent model (Xiao et al., 2018) trained on the ADE20K scene dataset (Zhou et al., 2017). The model can segment the input image into 336 object classes, 29 parts of large objects, and 25 materials. To further identify units that specialize in object parts, we expand each object class $c$ into additional object part classes $c$-$t$, $c$-$b$, $c$-$l$, and $c$-$r$, which denote the top, bottom, left, or right half of the bounding box of a connected component.

Below, we use dissection for analyzing and comparing units across datasets, layers, and models (Section 4.1), and locating artifact units (Section 4.2). Then, we start with a set of dominant object classes and use intervention to locate causal units that can remove and insert objects in different images (Section 4.3 and 4.4). In addition, our video demonstrates our interactive tool.

### 4.1 COMPARING UNITS ACROSS DATASETS, LAYERS, AND MODELS

**Emergence of individual unit object detectors**   We are particularly interested in any units that are correlated with instances of an object class with diverse visual appearances; these would suggest that GANs generate those objects using similar abstractions as humans. Figure 3 illustrates two such units. In the dining room dataset, a unit emerges to match dining table regions. More interestingly, the matched tables have different colors, materials, geometry, viewpoints, and levels of clutter: the only obvious commonality among these regions is the concept of a table. This unit's featuremap correlates to the fully supervised segmentation model (Xiao et al., 2018) with a high IoU of $0.34$.

**Interpretable units for different scene categories**   The set of all object classes matched by the units of a GAN provides a map of what a GAN has learned about the data. Figure 5 examines units from GANs trained on four LSUN scene categories (Yu et al., 2015). The units that emerge are object classes appropriate to the scene type: for example, when we examine a GAN trained on kitchen scenes, we find units that match stoves, cabinets, and the legs of tall kitchen stools. Another striking phenomenon is that many units represent parts of objects: for example, the conference room GAN contains separate units for the body and head of a person.

**Interpretable units for different network layers.**   In classifier networks, the type of information explicitly represented changes from layer to layer (Zeiler & Fergus, 2014). We find a similar phenomenon in a GAN. Figure 6 compares early, middle, and late layers of a progressive GAN with 14 internal convolutional layers. The output of the first convolutional layer, one step away from the input $z$, remains entangled: individual units do not correlate well with any object classes except for two units that are biased towards the ceiling of the room. Mid-level layers 4 to 7 have many units that match semantic objects and object parts. Units in layers 10 and beyond match local pixel patterns such as materials, edges and colors. All layers are shown in Section S-6.7.

**Interpretable units for different GAN models.**   Interpretable units can provide insights about how GAN architecture choices affect the structures learned inside a GAN. Figure 7 compares three models from Karras et al. (2018): a baseline Progressive GANs, a modification that introduces minibatch stddev statistics, and a further modification that adds pixelwise normalization. By examining unit semantics, we confirm that providing minibatch stddev statistics to the discriminator increases not only the realism of results, but also the diversity of concepts represented by units: the number of types of objects, parts, and materials matching units increases by more than $40\%$. The pixelwise normalization increases the number of units that match semantic classes by $19\%$.

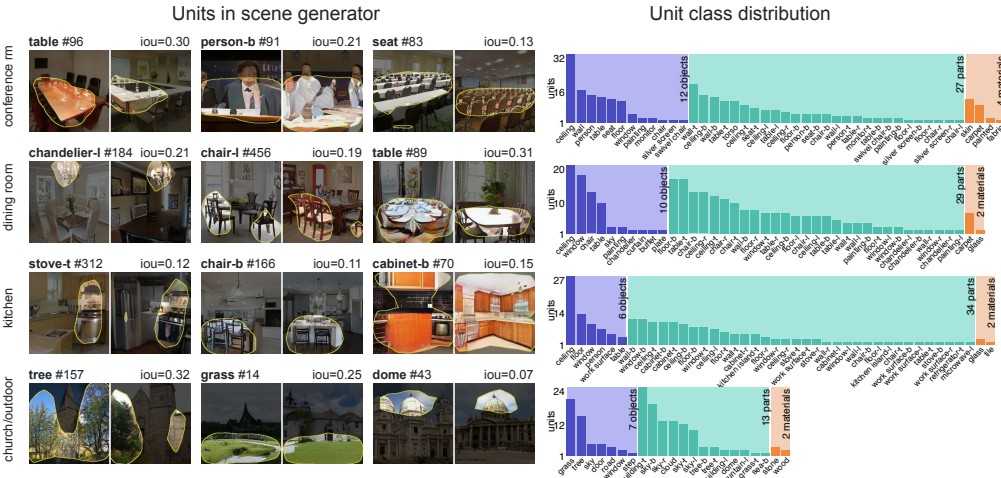

Figure 5: Comparing representations learned by progressive GANs trained on different scene types. The units that emerge match objects that commonly appear in the scene type: seats in conference rooms and stoves in kitchens. Units from `layer4` are shown. A unit is counted as a class predictor if it matches a supervised segmentation class with pixel accuracy $> 0.75$ and IoU $> 0.05$ when upsampled and thresholded. The distribution of units over classes is shown in the right column.

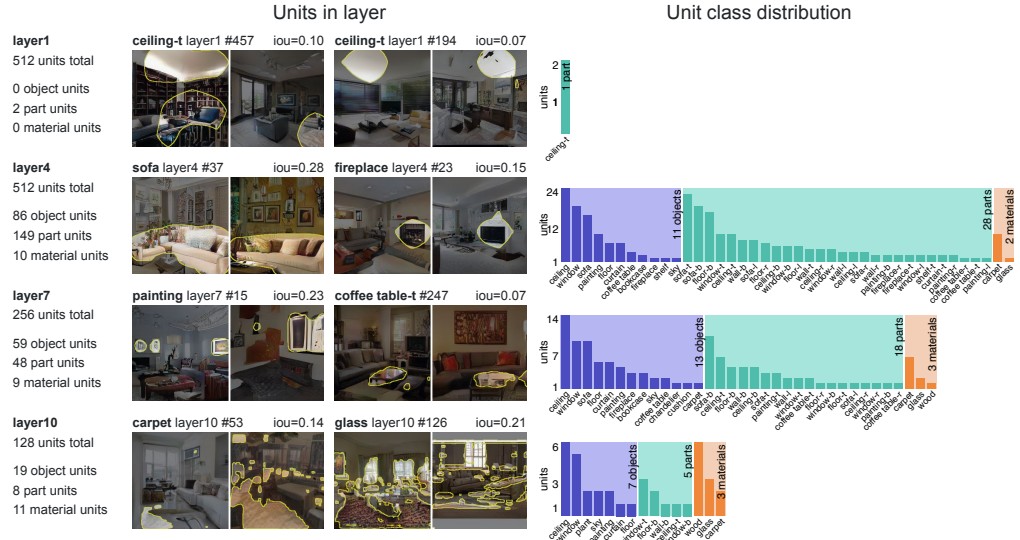

Figure 6: Comparing layers of a progressive GAN trained to generate LSUN living room images. The output of the first convolutional layer has almost no units that match semantic objects, but many objects emerge at layers 4-7. Later layers are dominated by low-level materials, edges and colors.

## 4.2 DIAGNOSING AND IMPROVING GANS

While our framework can reveal how GANs succeed in producing realistic images, it can also analyze the causes of failures in their results. Figure 8a shows several annotated units that are responsible for typical artifacts consistently appearing across different images. We can identify these units efficiently by human annotation: out of a sample of 1000 images, we visualize the top ten highest activating images for each unit, and we manually identify units with noticeable artifacts in this set. It typically takes 10 minutes to locate 20 artifact-causing units out of 512 units in `layer4`.

More importantly, we can fix these errors by ablating the above 20 artifact-causing units. Figure 8b shows that artifacts are successfully removed, and the artifact-free pixels stay the same, improving the generated results. In Table 1 we report two standard metrics, comparing our improved images to both the original artifact images and a simple baseline that ablates 20 randomly chosen units. First, we compute the widely used Fréchet Inception Distance (Heusel et al., 2017) between the generated images and real images. We use $50,000$ real images and generate $10,000$ images with high

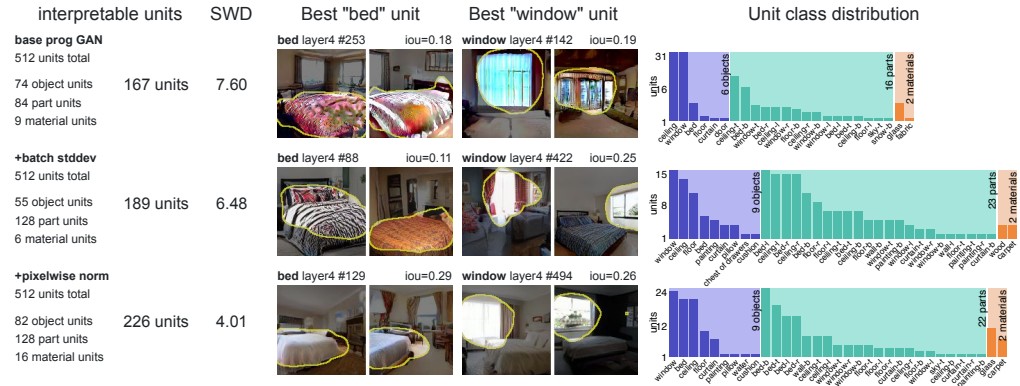

Figure 7: Comparing `layer4` representations learned by different training variations. Sliced Wasserstein Distance (SWD) is a GAN quality metric suggested by Karras et al. (2018): lower SWD indicates more realistic image statistics. Note that as the quality of the model improves, the number of interpretable units also rises. Progressive GANs apply several innovations including making the discriminator aware of minibatch statistics, and pixelwise normalization at each layer. We can see batch awareness increases the number of object classes matched by units, and pixel norm (applied in addition to batch stddev) increases the number of units matching objects.

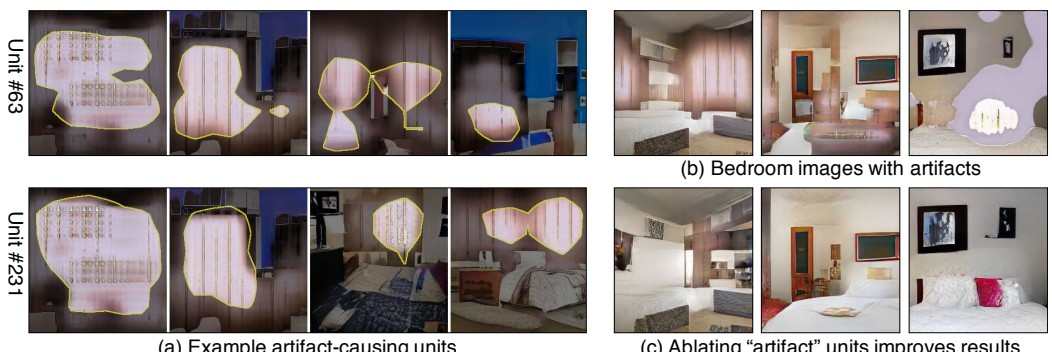

Figure 8: (a) We show two example units that are responsible for visual artifacts in GAN results. There are 20 units in total. By ablating these units, we can fix the artifacts in (b) and significantly improve the visual quality as shown in (c).

activations on these units. Second, we score $1,000$ images per method on Amazon MTurk, collecting $20,000$ human annotations regarding whether the modified image looks more realistic compared to the original. Both metrics show significant improvements. Strikingly, this simple manual change to a network beats state-of-the-art GANs models. The manual identification of "artifact" units can be approximated by an automatic scoring of the realism of each unit, as detailed in Section S-6.1.

## 4.3 LOCATING CAUSAL UNITS WITH ABLATION

Errors are not the only type of output that can be affected by directly intervening in a GAN. A variety of specific object types can also be removed from GAN output by ablating a set of units in a GAN. In Figure 9 we apply the method in Section 3.2 to identify sets of 20 units that have causal effects on common object classes in conference rooms scenes. We find that, by turning off these small sets of units, most of the output of people, curtains, and windows can be removed from the generated scenes. However, not every object can be erased: tables and chairs cannot be removed. Ablating those units will reduce the size and density of these objects, but will rarely eliminate them.

The ease of object removal depends on the scene type. Figure 10 shows that, while windows can be removed well from conference rooms, they are more difficult to remove from other scenes. In particular, windows are just as difficult to remove from a bedroom as tables and chairs from a conference room. We hypothesize that the difficulty of removal reflects the level of choice that a GAN has learned for a concept: a conference room is defined by the presence of chairs, so they cannot be altered. And modern building codes mandate that all bedrooms must have windows; the GAN seems to have caught on to that pattern.

Table 1: We compare generated images before and after ablating 20 "artifacts" units. We also report a simple baseline that ablates 20 randomly chosen units.

| Fréchet Inception Distance (FID) | |
| --- | --- |
| original images | 43.16 |
| "artifacts" units ablated (ours) | **27.14** |
| random units ablated | 43.17 |

| Human preference score | original images |
| --- | --- |
| "artifacts" units ablated (ours) | **72.4**% |
| random units ablated | 49.9% |

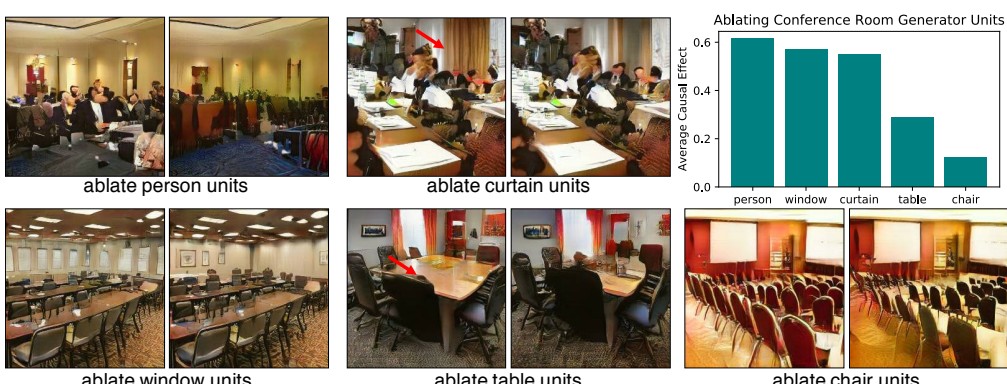

Figure 9: Measuring the effect of ablating units in a GAN trained on conference room images. Five different sets of units have been ablated related to a specific object class. In each case, 20 (out of 512) units are ablated from the same GAN model. The 20 units are specific to the object class and independent of the image. The average causal effect is reported as the portion of pixels that are removed in 1 000 randomly generated images. We observe that some object classes are easier to remove cleanly than others: a small ablation can erase most pixels for people, curtains, and windows, whereas a similar ablation for tables and chairs only reduces object sizes without deleting them.

## 4.4 CHARACTERIZING CONTEXTUAL RELATIONSHIPS VIA INSERTION

We can also learn about the operation of a GAN by forcing units on and inserting these features into specific locations in scenes. Figure 11 shows the effect of inserting 20 `layer4` causal door units in church scenes. In this experiment, we insert these units by setting their activation to the fixed mean value for doors (further details in Section S-6.4). Although this intervention is the same in each case, the effects vary widely depending on the objects' surrounding context. For example, the doors added to the five buildings in Figure 11 appear with a diversity of visual attributes, each with an orientation, size, material, and style that matches the building.

We also observe that doors cannot be added in most locations. The locations where a door can be added are highlighted by a yellow box. The bar chart in Figure 11 shows average causal effects of insertions of door units, conditioned on the background object class at the location of the intervention. We find that the GAN allows doors to be added in buildings, particularly in plausible locations such as where a window is present, or where bricks are present. Conversely, it is not possible to trigger a door in the sky or on trees. Interventions provide insight on how a GAN enforces relationships between objects. Even if we try to add a door in `layer4`, that choice can be vetoed later if the object is not appropriate for the context. These downstream effects are further explored in Section S-6.5.

## 5 DISCUSSION

By carefully examining representation units, we have found that many parts of GAN representations can be interpreted, not only as signals that correlate with object concepts but as variables that have a causal effect on the synthesis of objects in the output. These interpretable effects can be used to compare, debug, modify, and reason about a GAN model. Our method can be potentially applied to other generative models such as VAEs (Kingma & Welling, 2014) and RealNVP (Dinh et al., 2017).

We have focused on the generator rather than the discriminator (as did in Radford et al. (2016)) because the generator must represent all the information necessary to approximate the target distribution, while the discriminator only learns to capture the difference between real and fake images. Alternatively, we

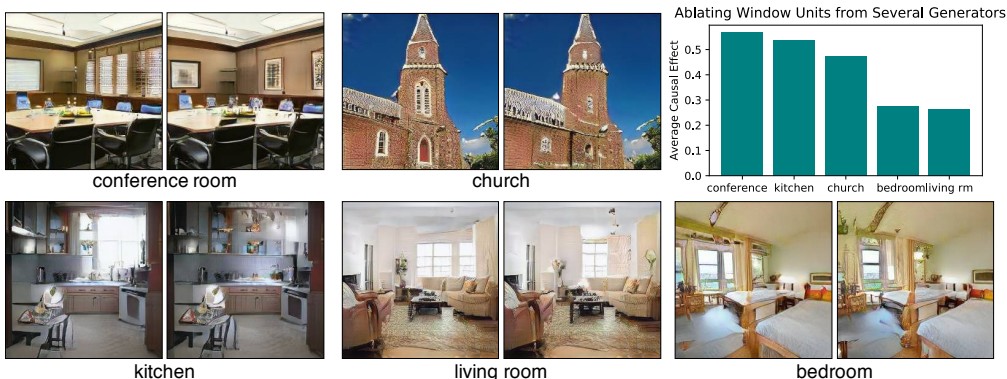

Figure 10: Comparing the effect of ablating 20 window-causal units in GANs trained on five scene categories. In each case, the 20 ablated units are specific to the class and the generator and independent of the image. In some scenes, windows are reduced in size or number rather than eliminated, or replaced by visually similar objects such as paintings.

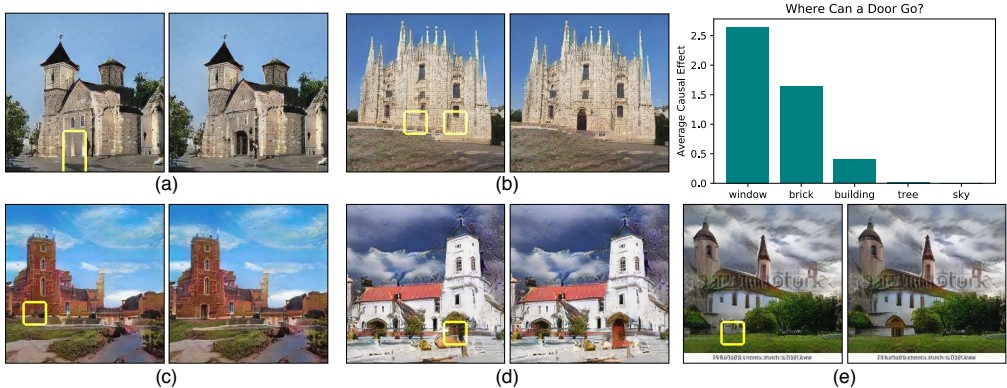

Figure 11: Inserting door units by setting 20 causal units to a fixed high value at one pixel in the representation. Whether the door units can cause the generation of doors is dependent on its local context: we highlight every location that is responsive to insertions of door units on top of the original image, including two separate locations in (b) (we intervene at left). The same units are inserted in every case, but the door that appears has a size, alignment, and color appropriate to the location. Emphasizing a door that is already present results in a larger door (d). The chart summarizes the causal effect of inserting door units at one pixel with different contexts.

can train an encoder to invert the generator (Donahue et al., 2017; Dumoulin et al., 2017). However, this incurs additional complexity and errors. Many GANs also do not have an encoder.

Our method is not designed to compare the quality of GANs to one another, and it is not intended as a replacement for well-studied GAN metrics such as FID, which estimate realism by measuring the distance between the generated distribution of images and the true distribution (Borji (2018) surveys these methods). Instead, our goal has been to identify the interpretable structure and provide a window into the internal mechanisms of a GAN.

Prior visualization methods (Zeiler & Fergus, 2014; Bau et al., 2017; Karpathy et al., 2016) have brought new insights into CNN and RNN research. Motivated by that, in this work we have taken a small step towards understanding the internal representations of a GAN, and we have uncovered many questions that we cannot yet answer with the current method. For example: why can a door not be inserted in the sky? How does the GAN suppress the signal in the later layers? Further work will be needed to understand the relationships between layers of a GAN. Nevertheless, we hope that our work can help researchers and practitioners better analyze and develop their own GANs.

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

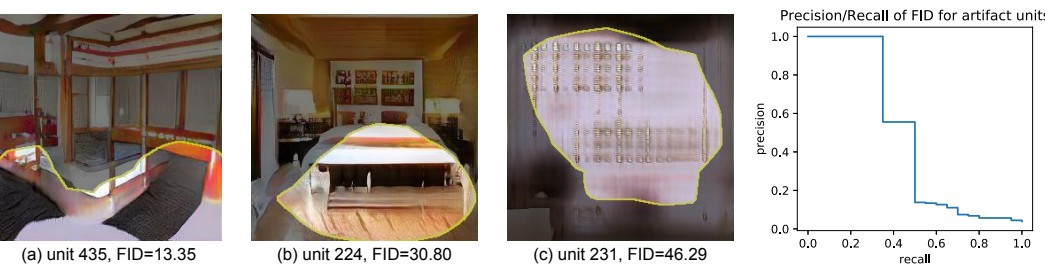

| (a) unit 435, FID=13.35 | (b) unit 224, FID=30.80 | (c) unit 231, FID=46.29 | |

Figure 12: At left, visualizations of the highest-activating image patches (from a sample of 1000) for three units. (a) the lowest-FID unit that is manually flagged as showing artifacts (b) the highest-FID unit that is not manually flagged (c) the highest-FID unit overall, which is also manually flagged. At right, the precision-recall curve for unit FID as a predictor of the manually flagged artifact units. A FID threshold selecting the top 20 FID units will identify 10 (of 20) of the manually flagged units.

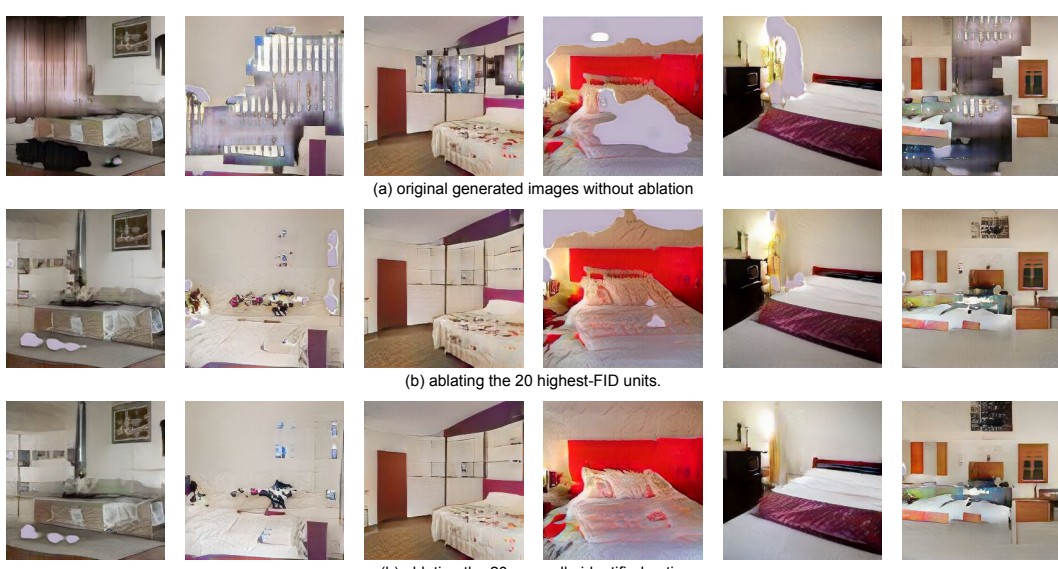

Figure 13: The effects of ablating high-FID units compared to manually-flagged units: (a) generated images with artifacts, without intervention; (b) those images generated after ablating the 20-highest FID units; (c) those images generated after ablating the 20 manually-chosen artifact units.

## S-6 SUPPLEMENTARY MATERIAL

### S-6.1 AUTOMATIC IDENTIFICATION OF ARTIFACT UNITS

In Section 4.2, we have improved GANs by manually identifying and ablating artifact-causing units. Now we describe an automatic procedure to identify artifact units using unit-specific FID scores.

To compute the FID score (Heusel et al., 2017) for a unit $u$, we generate $200,000$ images and select the $10,000$ images that maximize the activation of unit $u$, and this subset of $10,000$ images is compared to the true distribution ($50,000$ real images) using FID. Although every such unit-maximizing subset of images represents a skewed distribution, we find that the per-unit FID scores fall in a wide range, with most units scoring well in FID while a few units stand out with bad FID scores: many of them were also manually flagged by humans, as they tend to activate on images with clear visible artifacts.

Figure 12 shows the performance of FID scores as a predictor of manually flagged artifact units. The per-unit FID scores can achieve 50% precision and 50% recall. That is, of the 20 worst-FID units, 10 are also among the 20 units manually judged to have the most noticeable artifacts. Furthermore, repairing the model by ablating the highest-FID units works: qualitative results are shown in Figure 13 and quantitative results are shown in Table 2.

Table 2: We compare generated images before and after ablating "artifact" units. The "artifacts" units are found either manually, automatically, or both. We also report a simple baseline that ablates 20 randomly chosen units.

| Fréchet Inception Distance (FID) | |
|---|---|
| original images | 43.16 |
| manually chosen "artifact" units ablated (as in Section 4.2) | **27.14** |
| highest-20 FID units ablated | 27.6 |
| union of manual and highest FID (30 total) units ablated | 26.1 |
| 20 random units ablated | 43.17 |

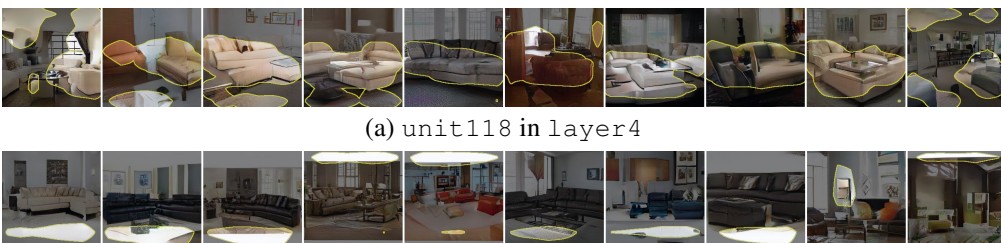

(a) `unit118` in `layer4`

(b) `unit11` in `layer4`

Figure 14: Two examples of generator units that our dissection method labels differently from humans. Both units are taken from `layer4` of a Progressive GAN of living room model. In (a), human label the unit as 'sofa' based on viewing the top-20 activating images, and our method labels as 'ceiling'. In this case, our method counts many ceiling activations in a sample of 1000 images beyond the top 20. In (b), the dissection method has no confident label prediction even though the unit consistently triggers on white letterbox shapes at the top and bottom of the image. The segmentation model we use has no label for such abstract shapes.

## S-6.2 HUMAN EVALUATION OF DISSECTION

As a sanity check, we evaluate the gap between human labeling of object concepts correlated with units and our automatic segmentation-based labeling, for one model, as follows.

For each of 512 units of `layer4` of a "living room" Progressive GAN, 5 to 9 human annotations were collected (3728 labels in total). In each case, an AMT worker is asked to provide one or two words describing the highlighted patches in a set of top-activating images for a unit. Of the 512 units, 201 units were described by the same consistent word (such as "sofa", "fireplace" or "wicker") in 50% or more of the human labels. These units are interpretable to humans.

Applying our segmentation-based dissection method, 154/201 of these units are also labeled with a confident label with IoU > 0.05 by dissection. In 104/154 cases, the segmentation-based model gave the same label word as the human annotators, and most others are slight shifts in specificity. For example, the segmentation labels "ottoman" or "curtain" or "painting" when a person labels "sofa" or "window" or "picture," respectively. A second AMT evaluation was done to rate the accuracy of both segmentation-derived and human-derived labels. Human-derived labels scored 100% (of the 201 human-labeled units, all of the labels were rated as consistent by most raters). Of the 154 segmentation-generated labels, 149 (96%) were rated by most AMT raters as accurate as well.

The five failure cases (where the segmentation is confident but rated as inaccurate by humans) arise from situations in which human evaluators saw one concept after observing only 20 top-activating images, while the algorithm, in evaluating 1000 images, counted a different concept as dominant. Figure 14a shows one example: in the top images, mostly sofas are highlighted and few ceilings, whereas in the larger sample, mostly ceilings are triggered.

There are also 47/201 cases where the segmenter is not confident while humans have consensus. Some of these are due to missing concepts in the segmenter. Figure 14b shows a typical example, where a unit is devoted to letterboxing (white stripes at the top and bottom of images), but the segmentation has no confident label to assign to these. We expect that as future semantic segmentation models are developed to be able to identify more concepts such as abstract shapes, more of these units can be automatically identified.

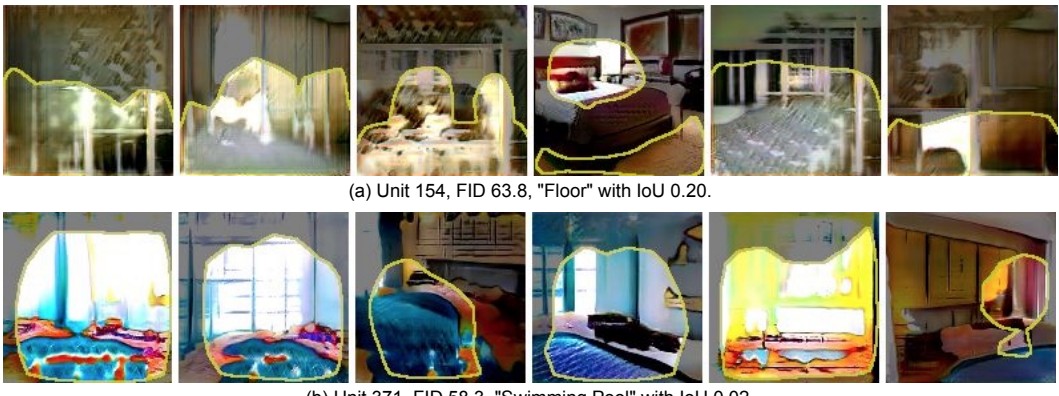

(a) Unit 154, FID 63.8, "Floor" with IoU 0.20.

(b) Unit 371, FID 58.3, "Swimming Pool" with IoU 0.02.

Figure 15: Two examples of units that correlate with unrealistic images that confuse a semantic segmentation network. Both units are taken from a WGAN-GP for LSUN bedrooms.

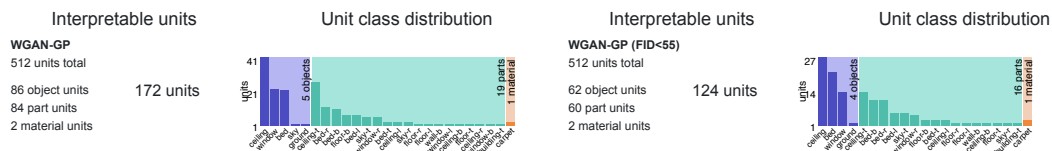

Figure 16: Comparing a dissection of units for a WGAN-GP trained on LSUN bedrooms, considering all units (at left) and considering only "realistic" units with FID < 55 (at right). Filtering units by FID scores removes spurious detected concepts such as 'sky', 'ground', and 'building'.

### S-6.3   PROTECTING SEGMENTATION MODEL AGAINST UNREALISTIC IMAGES

Our method relies on having a segmentation function $s_c(\mathbf{x})$ that identifies pixels of class $c$ in the output $\mathbf{x}$. However, the segmentation model $s_c$ can perform poorly in the cases where $\mathbf{x}$ does not resemble the original training set of $s_c$. This phenomenon is visible when analyzing earlier GAN models. For example, Figure 15 visualizes two units from a WGAN-GP model (Gulrajani et al., 2017) for LSUN bedrooms (this model was trained by Karras et al. (2018) as a baseline in the original paper). For these two units, the segmentation network seems to be confused by the distorted images.

To protect against such spurious segmentation labels, we can use a technique similar to that described in Section S-6.1: automatically identify units that produce unrealistic images, and omit those "unrealistic" units from semantic segmentation. An appropriate threshold to apply will depend on the distribution being modeled: in Figure 16, we show how applying a filter, ignoring segmentation on units with FID 55 or higher, affects the analysis of this base WGAN model. In general, fewer irrelevant labels are associated with units.

### S-6.4   COMPUTING CAUSAL UNITS

In this section we provide more details about the ACE optimization described in Section 3.2.

**Specifying the per-class positive intervention constant k.**   In Eqn. 3, the negative intervention is defined as zeroing the intervened units, and a positive intervention is defined as setting the intervened units to some big class-specific constant $\mathbf{k}$. For interventions for class $c$, we set $\mathbf{k}$ to be mean featuremap activation conditioned on the presence of class $c$ at that location in the output, with each pixel weighted by the portion of the featuremap locations that are covered by the class $c$. Setting all units at a pixel to $\mathbf{k}$ will tend to strongly cause the target class. The goal of the optimization is to find the subset of units that is causal for $c$.

**Sampling $c$-relevant locations P.**   When optimizing the causal objective (Eqn. 5), the intervention locations P are sampled from individual featuremap locations. When the class $c$ is rare, most featuremap locations are uninformative: for example, when class $c$ is a door in church scenes, most regions of the sky, grass, and trees are locations where doors will not appear. Therefore, we focus the optimization as follows: during training, minibatches are formed by sampling locations P that are

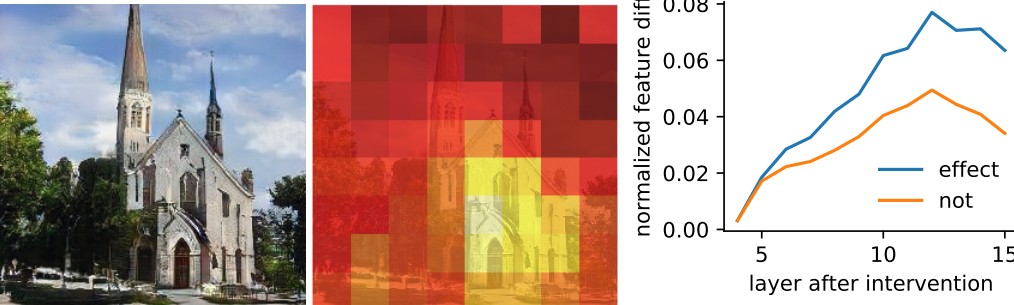

Figure 17: Tracing the effect of inserting door units on downstream layers. An identical "door" intervention at `layer4` of each pixel in the featuremap has a different effect on later feature layers, depending on the location of the intervention. In the heatmap, brighter colors indicate a stronger effect on the `layer14` feature. A request for a door has a larger effect in locations of a building, and a smaller effect near trees and sky. At right, the magnitude of feature effects at every layer is shown, measured by the changes of mean-normalized features. In the line plot, feature changes for interventions that result in human-visible changes are separated from interventions that do not result in noticeable changes in the output.

relevant to class $c$ by including locations where the class $c$ is present in the output (and are therefore candidates for removal by ablating a subset of units), and an equal portion of locations where class $c$ is not present at P, but it would be present if all the units are set to the constant $\mathbf{k}$ (candidate locations for insertion with a subset of units). During the evaluation, causal effects are evaluated using uniform samples: the region P is set to the entire image when measuring ablations, and to uniformly sampled pixels P when measuring single-pixel insertions.

**Initializing $\boldsymbol{\alpha}$ with IoU.**   When optimizing causal $\boldsymbol{\alpha}$ for class $c$, we initialize with

$$\alpha_u = \frac{\text{IoU}_{u,c}}{\max_v \text{IoU}_{v,c}} \tag{7}$$

That is, we set the initial $\alpha$ so that the largest component corresponds to the unit with the largest IoU for class $c$, and we normalize the components so that this largest component is 1.

**Applying a learned intervention $\boldsymbol{\alpha}$**   When applying the interventions, we clip $\boldsymbol{\alpha}$ by keeping only its top $n$ components and zeroing the remainder. To compare the interventions of different classes an different models on an equal basis, we examine interventions where we set $n = 20$.

### S-6.5   TRACING THE EFFECT OF AN INTERVENTION

To investigate the mechanism for suppressing the visible effects of some interventions seen in Section 4.4, in this section we insert 20 door-causal units on a sample of individual featuremap locations at `layer4` and measure the changes caused in later layers.

To quantify effects on downstream features, the change in each feature channel is normalized by that channel's mean L1 magnitude, and we examine the mean change in these normalized featuremaps at each layer. In Figure 17, these effects that propagate to `layer14` are visualized as a heatmap: brighter colors indicate a stronger effect on the final feature layer when the door intervention is in the neighborhood of a building instead of trees or sky. Furthermore, we plot the average effect on every layer at right in Figure 17, separating interventions that have a visible effect from those that do not. A small identical intervention at `layer4` is amplified to larger changes up to a peak at `layer12`.

### S-6.6   MONITORING GAN UNITS DURING TRAINING

Dissection can also be used to monitor the progress of training by quantifying the emergence, diversity, and quality of interpretable units. For example, in Figure 18 we show dissections of `layer4` representations of a Progressive GAN model trained on bedrooms, captured at a sequence of checkpoints during training. As training proceeds, the number of units matching objects increases, as does the number of object classes with matching units, and the quality of object detectors as

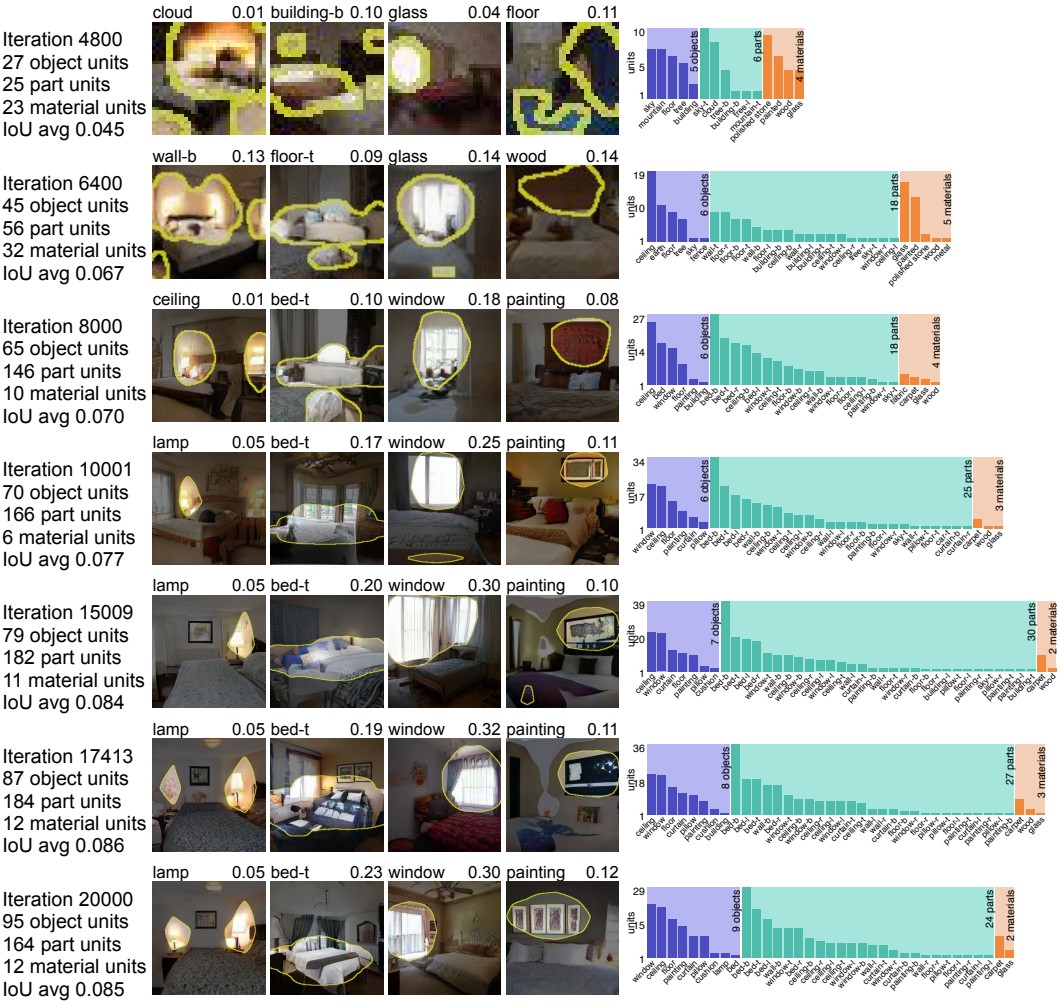

Figure 18: The evolution of `layer4` of a Progressive GAN bedroom generator as training proceeds. The number and quality of interpretable units increases during training. Note that in early iterations, Progressive GAN generates images at a low resolution. The top-activating images for the same four selected units is shown for each iteration, along with the IoU and the matched concept for each unit at that checkpoint.

measured by average IoU over units increases. During this successful training, dissection suggests that the model is gradually learning the structure of a bedroom, as increasingly units converge to meaningful bedroom concepts.

## S-6.7 ALL LAYERS OF A GAN

In Section 4.1 we show a small selection of layers of a GAN; in Figure 19 we show a complete listing of all the internal convolutional layers of that model (a Progressive GAN trained on LSUN living room images). As can be seen, the diversity of units matching high-level object concepts peaks at `layer4-layer6`, then declines in later layers, with the later layers dominated by textures, colors, and shapes.

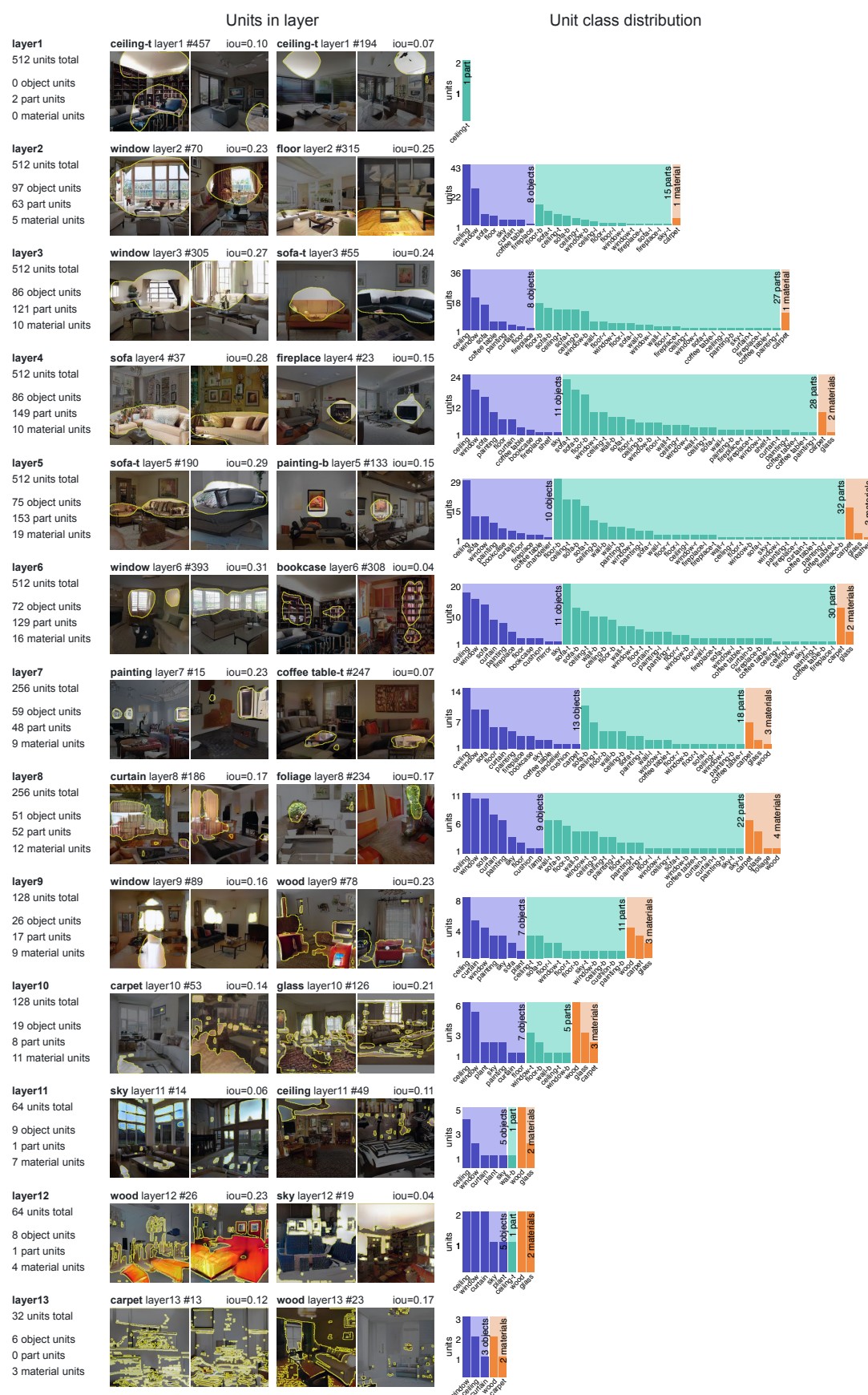

Figure 19: All layers of a Progressive GAN trained to generate LSUN living room images.

