# OpenReview forum: "GAN Dissection: Visualizing and Understanding Generative Adversarial Networks"
_ICLR.cc/2019/Conference_

### Official Review · AnonReviewer1 · 2018-11-02
**This paper reveals the essence of GAN through experiments.**

**Rating:** 8
**Confidence:** 4

**Review:**

This paper provides a visualization framework to understand the generative neural network in GAN models. To achieve this, they first find a group of interpretable units and then quantify the causal effect of interpretable units. Finally, the contextual relationship between these units and their surrounding is examined by inserting the discovered object concepts into new images. Extensive experiments are presented and a video is provided.

Overall, I think this paper is very valuable and well-written. The experiments clearly show the questions proposed in the introduction are answered. Two concerns are as follows.

Cons:
1) The visualization seems to be very heuristic. What I want to know is the theoretical interpretation of the visualization. For example, the Class Activation Maps (CAM) can be directly calculated by the output values of softmax function. How about the visual class for the generative neural networks?
2) I am also very curious, how is the rate of finding the correct sets of units for a particular visual class?

---

> ### Author Response · Authors · 2018-11-26
> **Answers to questions for AnonReviewer1**
>
> Thank you for your comments and questions; we have incorporated your suggestions in the revision, and we also answer your questions below.
>
> Q1: Theoretical interpretation of the visualization, and comparisons to the Class Activation Maps (CAM)?
>
> A1: Our visualization is very simple and corresponds to equation (2): we upsample a single channel of the activation featuremap and show the region exceeding a threshold: unlike CAM, no gradients are considered. The threshold used is chosen to maximize relative mutual information with the best-matching object class based on semantic segmentation, however, a fixed threshold such as a top 1% quantile level would look very similar.
>
> It is also informative to consider a CAM-like visualization of the causal impact of interventions in the model on later layers: we can create a heatmap where each pixel shows the magnitude of the last featuremap layer change that results when making an intervention at each pixel in an early layer.  The result is shown in Figure 17 of supplementary materials S-6.4: this visualization shows that the effects of an intervention at different locations are not uniform. The heatmap pattern reveals the structure of the model’s sensitivity to a specific concept at various locations.
>
> Q2: How is the rate of finding the correct sets of units for a particular visual class?
>
> A2:   Our method provides a correct label for 96% of interpretable units, as measured by the following human evaluation, which we have added to supplementary materials, section S-6.2.
>
> For each of 512 units of layer 4 of a "living room" progressive GAN,  5-9 human labels are collected (3728 labels total), where the AMT worker is asked to provide one or two words describing the highlighted patches in a set of top-activating images for a unit.  Of the 512 units, 201 units were described by a consistent word (such as "sofa", "fireplace" or "wicker") that was supplied by 50% or more of the human labels.
>
> Applying our segmentation-based dissection method, 154/201 of these units are also labeled with a confident label with IoU > 0.05 by dissection.  In most of the cases (104/154), the segmentation-based method gave the same label word as the human labelers, and most others are slight shifts in specificity (e.g. segmentation says "ottoman" or "curtain" or "painting" when a person says "sofa" or "window" or "picture").  A second AMT evaluation was done to rate the accuracy of both segmentation-derived and human-derived labels.  Human-derived labels scored 100% (i.e., of the 201 human-labeled units, all of the labels were rated to be accurate by most raters).  Of the 154 of our segmentation-generated labels, 149 (96%) were rated as accurate by most AMT raters as well.
>
> The five failure cases (where the segmentation is confident but rated as inaccurate by humans) arise from situations in which human evaluators saw one pattern from seeing only 20 top-activating images, while the algorithm, in evaluating 1000 images, counted a different concept as dominant.  (E.g., in one example shown in Figure 14a, there are only a few ceilings highlighted and mostly sofas, whereas in the larger 1000-image set, mostly ceilings are triggered.)
>
> There were also 47/201 cases where the segmenter was not confident while humans had consensus.  Some of these are due to missing concepts in the segmenter.  For example, several units are devoted to letterboxing (white stripes at the top and bottom of images), and the segmentation had no confident label to assign to these (Figure 14b).
>
> We expect that as semantic segmentations improve to be able to identify more concepts such as abstract shapes, more of these units can be automatically identified.

---

### Official Review · AnonReviewer2 · 2018-11-03
**An interesting idea to visualize and explain the representation of GANs and to provide a new potential way to further improve the quality of the generated images by GANs**

**Rating:** 7
**Confidence:** 4

**Review:**

## Summary
This work proposes a novel analytic framework exploited on a semantic segmentation model to visualize GANs at unit (feature map) level. The authors show that some GAN representations can be interpreted, correlate with the parsing result from the semantic segmentation model but as variables that have a causal effect on the synthesis of semantic objects in the output. This framework could allow to detect and remove the artifacts to improve the quality of the generated images.

The paper is well-written and organized. The dissection and intervention for finding relationships between representation units and objects are simple, straightforward and meaningful. The visualizations are convincing and insightful. I recommend to accept the paper.

## Detail comments
About diagnosing and improving GANs, please give more details of the human annotation for the artifacts.

I think there is a typo in the first and second paragraphs in section 4.3, Figure 14 -> Figure 8.

The whole framework is based on a semantic segmentation model. The model is highly possibly imperfect and could have very different performances on different objects. Have you ever considerate to handle these imperfect models?

Is there a way to apply the framework to the training process of GANs?

---

> ### Author Response · Authors · 2018-11-26
> **Answers to questions for AnonReviewer2**
>
> Thank you for your comments and questions; we have incorporated your suggestions in the revision, and we answer your questions below.
>
> Q3: About diagnosing and improving GANs, please give more details of the human annotation for the artifacts.
>
> A3: We visualize the top 10 highest activating images for each unit, and we manually identify units with noticeable artifacts in this set.  (This human annotation was done by an author.)
> Details have been added to section 4.2.  This method for diagnosing and improving GANs is further analyzed and expanded in the supplementary materials, in section S-6.1.
>
> Q4: Minor - I think there is a typo in the first and second paragraphs in section 4.2, Figure 14 -> Figure 8.
>
> A4: Thanks for your detailed comments. We have fixed it.
>
> Q5: Have you ever considered to handle these imperfect semantic segmentation models?
>
> A5: We totally agree with the reviewer: the success of our method is linked to the accuracy and comprehensiveness of the segmentation model used.  We have performed a human evaluation regarding the accuracy of our method on a Progressive GAN model (on LSUN living rooms), and have found that, our method provides correct labels for 96% of interpretable units.  Further details of the evaluation can be found in section S-6.2.
>
> In addition, a semantic segmentation model can perform poorly if the analyzed images are very different from the images on which the semantic segmentation was trained.  For example in the “bedroom” scene category, if a unit is labeled as correlating with ‘swimming pool’ this may be due to a poorly performing GAN model. We have partly addressed this issue by measuring the average realism of each unit using the FID metric. In practice, in Figure 16, we show the effect of such a filter in which we only report “realistic” and interpretable units.  Details of such an approach have been added to section S-6.3.
>
> As more accurate and robust segmentation models are developed, we expect our method to be able to identify more semantic concepts inside a representation.
>
>
> Q6: Is there a way to apply the framework to the training process of GANs?
>
> A6: By using a per-unit realism score based on the FID metric on generator units learned by the GAN, we can identify units that should be zeroed to improve the realism of the GAN output.  (We assign a realism score to each unit by measuring FID for a subset of images that highly activate the unit.) Zeroing the units with the highest FID score as measured this way will improve the quality of the output nearly as well as ablating units identified manually. This modification could be incorporated into an automatic training process. S-6.1 has further details and a preliminary evaluation of this idea for introducing per-unit analysis in an automatic process.  A full development of this idea is left to future work.
>
> Dissection can also be used to monitor the progress of training by quantifying the emergence, diversity, and quality of semantic units.  For example, in Figure 18 we show dissections of layer4 representations of a Progressive GAN model trained on bedrooms, captured at a sequence of checkpoints during training.  As training proceeds, the number of units matching objects (and the number of object classes with matching units) increases, and the quality of object detectors as measured by average IoU over units increases.  During this successful training, dissection suggests that the model is learning the structure of a bedroom, because increasingly units converge to meaningful bedroom concepts.  We add this analysis to section S-6.6.

---

### Official Review · AnonReviewer3 · 2018-11-04
**New methods for interpreting GANs, with nice practical contribution for improving GANs outputs.**

**Rating:** 7
**Confidence:** 3

**Review:**

The paper proposes a method for visualizing and understanding GANs representation. This seems an important topic as several such methods were performed for networks trained in supervised learning, which relate
to the predicted outcome, but there is lack of methods for interpreting GANs which are learned in an unsupervised manner and it is generally unclear what is the representation learned by GANs.
The method is finding correlations between the appearance of objects and the activation of units in each layer of the learned network.
In addition, the paper presents a 'causal' measure, where a causal effect of a unit is measured by removing and adding this unit from/to the network and computing the average effect on object appearance.
The authors demonstrate how the methods are applied by improving the appearance of images, by modifying units which were detected as important for specific objects.
The authors also provide an interactive interface where users can manually examine and modify their trained GANs in order to add/remove objects and to remove artifacts.

The method proposed by the authors seem to be appropriate for convolutional neural networks, where 'units' in each layer may correspond to objects and can be searched for in particular locations of image.
It is not clear to me if and how one can apply the author's methods to other architecture, and to other application domains (besides images), or whether the method is limited to vision applications.
The authors do not explain specifically how do they choose the 'units' for which they seek interpretation when reporting their results. It is written that each layer is divided into two sets:
u  and u-bar, where we seek interpretation of u. But how large does u tend to be? how would one choose it? is it one filter out of all filters in a certain layer? when optimizing for sets of units together
(using the alpha probabilities and the optimization in eq. 6) what is d? is it performed for all units in a single layer? more details would be useful here.

The paper is overall clearly written, with lots of visual examples demonstrating the methods presented in it.
The paper presents a new methodological idea, which allows for nice practical contribution. There is no theoretical contribution or any deep analysis.
There is no reference in the paper to the supp. info. figures and therefore it is not clear if and how the supp. info. adds valuable information to the reader.
The authors use scores like SWD and FIT for performance, but give no explanations for what do these scores measure.


Minor:

Abstract: immprovements -> improvements

Page 6, middle: 'train on four LSUN' -> 'trained on four LSUN'

Page 7, bottom: Fig. 14a and 14b should be Fig. 8a and 8b

---

> ### Author Response · Authors · 2018-11-26
> **Answers to questions for AnonReviewer3**
>
> Thank you for your comments and questions; we have incorporated your suggestions in the revision, and we also answer your questions below.
>
> Q7:  apply the author's methods to other architecture, and to other application domains?
>
> A7: We have applied our method to WGAN-GP model with a different generator architecture, as shown in Figure 16 in Section S-6.3. Our method can find interpretable units for different GANs objectives and architectures.
>
> The general framework can be extended beyond generative models for vision, although that topic is beyond the scope of the current paper.  Concurrent work submitted to ICLR 2019 is an example of similar ideas being applied to natural language translation. (https://openreview.net/forum?id=H1z-PsR5KX)
>
> Q8: how to choose the 'units' for which they seek interpretation when reporting their results?
>
> A8: We do two analyses.  For the dissection analysis examining correlation, u are analyzed as individual units (i.e., |U| = 1). We analyze every individual unit in a layer, and we plot all units that match a segmented concept with IoU exceeding 5%.
>
> For the causal analysis, we choose the elements of U by doing the optimization described in equation (6), which finds an alpha that specifies a contribution for every unit to maximize causal effects, ranking units according to highest alpha, and choosing the number needed to achieve a desired causal effect.
>
> Q9:  How large does u tend to be? How would one choose it?  Is it one filter out of all filters in a certain layer?
>
> A9: To choose U to have strong causal effects, we measure and plot the causal effect of different numbers of units for U as in Figure 4. The increase in causal effect diminishes after about 20 units. To be able to compare different causal sets on an equal basis, we set |U| = 20 for most of our experiments.
>
> Q10: When optimizing for sets of units together (using the alpha probabilities and the optimization in eq. 6) what is d? Is it performed for all units in a single layer? More details would be useful here.
>
> A10: Yes, we perform an optimization for all units in a single layer.  d is the number of all units in a single layer (512, for the case of layer 4 of our Progressive GAN).
>
> For the dissection analysis, we analyze every individual unit in a layer, and we plot all units that match a segmented concept with IoU exceeding 5%.  The causal analysis requires identifying sets of units, which is done through the optimization in equation (6).
>
> Beyond this objective, learning U involves several additional details including how to specify the big constant for positive intervention, how to sample class-relevant positions, and how to initialize the coefficient alpha. We have added a section S-6.4 to supplementary materials with these implementation details.
>
> Q11: Regarding SWD and FID
>
> A11: SWD and FID are measures which estimate realism of the GAN output by measuring the distance between the generated distribution of images and the true distribution of images; Borji (arXiv 2018) surveys and compares these methods at https://arxiv.org/abs/1802.03446.  We have clarified these terms and added citations in the paper.
>
> Q12: No reference to supp. info and minor typos:
>
> A12: Thank you for your detailed comments; we have updated the text and expanded the supplementary materials. We also added a brief summary of the supplementary material in each section of the main paper.

---

### Author Response · Authors · 2018-11-26
**Summary of changes to the manuscript**

We thank all the reviewers for their helpful comments. We are glad that they found the topic important, the idea new, and the visualization results convincing. We have addressed individual questions raised by the reviewers in separate posts. Below we summarize the major changes in this revision.

- In supplementary material S-6.1, we show an automatic evaluation of per-unit realism that can be done using FID measurements, and we show that zeroing these units improves the quality of the output. We have also corrected our FID computation by eliminating JPEG artifacts in our evaluation pipeline and recomputed FID comparisons in Table 1. (R2Q3, R2Q6)
- In S-6.2, we conduct a human evaluation of dissection label accuracy for interpretable units. (R1Q2, R2Q5)
- In S-6.3, we show how unit realism can be used to filter the results to protect the segmenter against unrealistic images that can be produced by some GAN models. (R2Q5, R3Q7)
- In S-6.4, we provide details of our method for optimizing causal units. To eliminate a hyperparameter, we have defined the large constant “c” used for positive interventions to be a mean conditioned on the target class, rather than an unconditional 99 percentile value.  Figures 4, 9, 10, and 11 have been updated with results based on this adjustment. (R3Q10)
- In S-6.5, we have traced the effects of interventions through downstream layers and show how a CAM-like heatmap can be used to visualize these effects. (R1Q1)
- In S-6.6, we show how dissection can be used to monitor the emergence of unit semantics during the training epochs of a GAN. (R2Q6)
- We have fixed minor typos and grammar errors (R2Q4, R3Q12)
- We have clarified the method for manually identifying artifact units (R2Q3)
- We have clarified the method for identifying causal sets of units described in equations 5 and 6 (R3Q8,9,10)
- We have clarified the definition of SWD and FID and added citations (R3Q11)

---

### Meta-Review · Area_Chair1 · 2018-12-14
**Intersting framework for the analysis of GANs**

**Confidence:** 4
**Recommendation:** Accept (Poster)

**Metareview:**

The paper proposes an interesting framework for visualizing and understanding GANs, that will be of clear help for understanding existing models and might provide insights for developing new ones.